# Preservation of Mitochondrial Health in Liver Ischemia/Reperfusion Injury

**DOI:** 10.3390/biomedicines11030948

**Published:** 2023-03-20

**Authors:** Ivo F. Machado, Carlos M. Palmeira, Anabela P. Rolo

**Affiliations:** 1CNC—Center for Neuroscience and Cell Biology, University of Coimbra, 3000 Coimbra, Portugal; 2IIIUC—Institute of Interdisciplinary Research, University of Coimbra, 3000 Coimbra, Portugal; 3Department of Life Sciences, University of Coimbra, 3000 Coimbra, Portugal

**Keywords:** liver, ischemia-reperfusion, mitochondria, steatosis, aging

## Abstract

Liver ischemia-reperfusion injury (LIRI) is a major cause of the development of complications in different clinical settings such as liver resection and liver transplantation. Damage arising from LIRI is a major risk factor for early graft rejection and is associated with higher morbidity and mortality after surgery. Although the mechanisms leading to the injury of parenchymal and non-parenchymal liver cells are not yet fully understood, mitochondrial dysfunction is recognized as a hallmark of LIRI that exacerbates cellular injury. Mitochondria play a major role in glucose metabolism, energy production, reactive oxygen species (ROS) signaling, calcium homeostasis and cell death. The diverse roles of mitochondria make it essential to preserve mitochondrial health in order to maintain cellular activity and liver integrity during liver ischemia/reperfusion (I/R). A growing body of studies suggest that protecting mitochondria by regulating mitochondrial biogenesis, fission/fusion and mitophagy during liver I/R ameliorates LIRI. Targeting mitochondria in conditions that exacerbate mitochondrial dysfunction, such as steatosis and aging, has been successful in decreasing their susceptibility to LIRI. Studying mitochondrial dysfunction will help understand the underlying mechanisms of cellular damage during LIRI which is important for the development of new therapeutic strategies aimed at improving patient outcomes. In this review, we highlight the progress made in recent years regarding the role of mitochondria in liver I/R and discuss the impact of liver conditions on LIRI.

## 1. Introduction

Hepatic malignancies, acute liver failure and end-stage liver disease are commonly treated by liver transplantation. In recent years, the success of liver transplantation has improved, but unfortunately, the number of patients on waiting lists keeps growing [1]. Current efforts are to increase the pool of liver donors, for instance, by including marginal liver donors [2]. Liver ischemia-reperfusion (I/R) injury (LIRI) contributes to organ shortage as healthy livers and livers from marginal donors are susceptible to this type of damage [3], which can eventually result in acute and chronic rejection.

The liver has a remarkable ability to regenerate following toxic or physical damage. Upon liver injury, complex physiological and cellular events take place to fully restore the lost hepatic mass while adequate hepatic function is maintained to preserve body homeostasis. This ability allows the liver to successfully recover from resection and transplantation. Generally, LIRI is comprised by two distinct phases. The initial phase starts with interruption of hepatic circulation. During this phase (ischemia), nutrient and oxygen deprivation, pH changes, and adenosine triphosphate (ATP) depletion together lead to the aberrant formation of reactive oxygen species (ROS) and eventually cellular injury starts to set in. In the following phase, reperfusion of the liver exacerbates the damage initiated during ischemia due to metabolic disturbances and induction of a proinflammatory immune response (reviewed in [4,5]).

Mitochondria are critical players in energy and glucose metabolism, as well as in the regulation of several signaling pathways. Dysfunction of mitochondria is tightly linked with many human pathologies and aging [6,7]. Maintenance of mitochondrial health is thus required to ensure cellular and body homeostasis. Mitochondrial biogenesis, mitochondrial fission/fusion, and mitophagy are the main mechanisms that guarantee proper mitochondrial function in adaption to multiple stresses. Impaired mitochondrial function is one of the main causes for liver damage following I/R, however, the underlying mechanisms are still not completely comprehended. Their full understanding is likely to instigate the development of novel strategies capable of improving the surgical outcome of liver transplantation and resection, as well as leading to the implementation of measures that could decrease the number of patients on waiting lists. The current state of pharmacological and surgical approaches utilized to improve LIRI has been discussed in depth elsewhere and will not be reviewed here [8]. In the current review, we summarize the mitochondrial mechanisms underlying LIRI and discuss how maintenance of mitochondrial quality control contributes to the amelioration of LIRI. The impact of hepatic steatosis and liver aging on I/R injury will also be topics of discussion.

## 2. Cellular and Molecular Mechanisms of LIRI

The liver is an essential organ in vertebrate animals that is responsible for a complexity of functions, including biotransformation of xenobiotics, regulation of metabolites and nutrients, and maintenance of body homeostasis. Loss of hepatic function can have a dramatic effect on the organism, and it can even be fatal [9,10]. As the main parenchymal liver cells, hepatocytes are responsible for conducting most liver functions, while liver non-parenchymal cells support hepatocytes’ function. These include liver sinusoidal endothelial cells (LSECs), biliary duct cells (cholangiocytes), hepatic stellate cells (HSCs), and Kupffer cells. LIRI arises from a complex and intertwined network events taking placing after an ischemic insult that involve the interaction between the different liver cell types. In broad strokes, I/R results in hepatocyte and LSEC death due to metabolic disturbances and oxidative damage. Aggravation of the initial ischemic injury by reperfusion leads to the development of a strong inflammatory immune response due to release of damage-associated molecular patterns (DAMPs) and proinflammatory cytokines, and activation of the complement system. In turn, activation of Kupffer cells, recruitment and adhesion of neutrophils, and platelet activation sustain the inflammatory immune response which exacerbates LIRI (reviewed in [5]). LIRI compromises liver repair mechanisms and impairs liver function which may result in organ failure.

Liver cells are arranged in functional structural units, lobules, that are repeating hexagonal-shaped histological units subdivided into three separate concentrical zones [11]. Each zone has different metabolic functions. Catabolic processes, such as gluconeogenesis and β-oxidation, are prevalent in the oxygen-enriched periportal zone (zone 1) that receives arterial blood through the portal triads, while anabolic processes, such as glycolysis and lipogenesis, are prevalent in the pericentral zone (zone 3) where non-oxygenated blood that has flown through the liver sinusoids is drained into the central vein. The mid-lobule zone (zone 2) is thought to be a transitional zone with no specific function. Hepatocytes from different zones may not only have distinct metabolic functions but also contribute differently to the repopulation of liver cells during liver homeostasis and liver regeneration after injury [12]. Each liver zone is differently affected by liver I/R injury. Compared with the periportal and mid-lobule zones, the pericentral zone is more sensitive to hepatic I/R injury [13,14,15]. Pericentral hepatocytes are thought to be more vulnerable to anoxia-induced injury than periportal hepatocytes, because zone 3 has a lower oxygen concentration [11,16]. Spatial transcriptomic analysis showed that zones 1 and 3 have different profiles following I/R. While zone 3 is characterized by differentially expressed genes related to the inflammatory response, unfolded protein response, autophagy and metabolic pathways, zone 1 is mainly enriched in metabolic pathways [17]. The authors also showed that macrophages are mainly recruited to zone 3 which is associated with a higher inflammatory response following I/R. The higher presence of KCs, HSCs, and epithelial cells in zone 1 may be related to greater protection against I/R injury [17].

Hepatocytes and LSECs are the most susceptible cells to liver I/R-induced death [18,19], and thus we will mainly focus on them. But their sensitivity to warm (37 °C) and cold (4 °C) ischemia is different. While warm ischemia is observed in clinical settings such as liver transplantation, liver resection, and trauma, cold ischemia is typically found in the setting of liver transplantation, from the moment when the liver is cold stored for transport until it is implanted into the recipient. During warm ischemia, the interruption of blood flow leads to anoxia and nutrient depletion causing the decline of ATP synthesis. Consequent disturbances in intracellular metabolic processes cause acidification of the cytosol due to lactate accumulation. Once the blood flow is restored, reoxygenation leads to an increase in the production of mitochondrial ROS, cells are overloaded with calcium ions (Ca^2+^), and the pH of the tissue returns to physiological levels [4,20]. These events trigger the opening of the mitochondrial permeability transition (MPT) pore resulting in the loss of mitochondrial membrane integrity and subsequent hepatocyte death. Opening of the MPT pore allows molecules whose molecular mass is inferior to 1.5 kDa to diffuse freely across the inner mitochondrial membrane (IMM) [21]. In physiological conditions, the transient opening of the MPT pore is linked to mitochondrial energy metabolism, regulation of ROS signaling and mitochondrial Ca^2+^ signaling [22]. However, during I/R, prolonged MPT pore opening results in the dissipation of the proton motive force, uncoupling of oxidative phosphorylation, and mitochondrial swelling, which, in turn, lead to apoptotic and necrotic cell death [22,23]. In parallel, damaged parenchymal liver cells release DAMPs such as nuclear DNA, histones, heat shock proteins, ATP, mitochondrial formyl peptide, and mitochondrial DNA (mtDNA) that activate an inflammatory immune response [4,5]. The proinflammatory response is initiated due to the activation of Kupffer cells by DAMPs. Once they become activated, they produce ROS and release proinflammatory cytokines and chemokines [24]. Consequently, additional increase in ROS production foments cellular damage and apoptosis. The release of cytokines and chemokines augments the inflammatory response after reperfusion due to recruitment of monocytes, neutrophils and T cells, resulting in the exacerbation of hepatocellular injury [24].

On the other hand, studies report that LSECs are most susceptible to injury during cold storage [19,25,26]. Changes to their physiology and function result in the impairment of their viability. The transcription factor Krüppel-like factor 2 (KLF2), typically expressed by LSECs, is implicated in vasodilation, and has anti-thrombotic and anti-inflammatory effects. During cold ischemia, KLF2 is downregulated alongside with endothelial nitric oxide synthase (eNOS), thrombomodulin, and nuclear factor erythroid 2-related factor 2 (Nrf2). Restoration of these genes in cold stored rat livers prevents liver damage [27]. LSECs also release DAMPs that contribute to the development of an immune response and express adhesive molecules that promote neutrophil binding [26].

## 3. Mitochondrial Function and Dynamics during Liver I/R

Mitochondria efficiently produce chemical energy (ATP) through oxidative phosphorylation in most eukaryotic cells. These organelles have a wide variety of important functions other than energy metabolism, including metabolism of lipids, amino acids and nucleotides, ROS signaling, calcium homeostasis, and apoptosis [28]. Perturbations to mitochondrial health have serious effects on whole-body homeostasis. Mitochondrial dysfunction has been reported in several diseases involving different organs, such as the liver. In fact, impaired mitochondrial function hinders liver function and contributes to many liver diseases, such as steatosis, non-alcoholic steatohepatitis (NASH), and diabetes [6,29]. As a central coordinator of body glucose and energy metabolism, the liver heavily relies on mitochondria to fully satisfy its functional needs. In the setting of liver I/R, excessive production of ROS and mitochondrial membrane permeabilization, which all result in impaired mitochondrial function, are leading causes of hepatocyte death and LIRI. Targeting mitochondria with antioxidants confers protection against oxidative stress arising from ROS and has hepatoprotective effects against LIRI. For example, *N*-acetylcysteine (NAC) has been reported to reduce liver injury after I/R injury by decreasing ROS levels, proinflammatory cytokines, and cell death [30,31]. Similarly, targeting mitochondria with coenzyme Q10 protects against LIRI by reducing oxidative stress [32,33].

A variety of mitochondrial quality control mechanisms are activated in response to stress for the preservation of a normal mitochondrial environment [34,35]. The adaptation of mitochondrial function to stress depends on the fine-tuned coordination between nuclear and mitochondrial genomes [36]. To fully optimize mitochondrial function and preserve cellular homeostasis, mitonuclear communication acts in parallel with mitochondrial quality control mechanisms, which include mitochondrial biogenesis, mitochondrial dynamics, and mitophagy. For instance, the generation of functional mitochondria through mitochondrial biogenesis implicates the transcription and translation of genes from both nuclear and mitochondrial genomes. The quantity and quality of mitochondria therefore relies upon the equilibrium between mitochondrial proliferation and degradation. Disrupting this balance results in the decline of cellular function and cell death, which is a feature found in several pathologies [37]. Deficient mitochondrial biogenesis results in mitochondrial stress and decreased mitochondrial mass, while impaired mitophagy leads to the build-up of defective mitochondria. The remodeling of the mitochondrial network is also complemented by processes of mitochondrial fission and fusion. While fusion allows functional mitochondria to join the mitochondrial network, fission is important to separate damaged mitochondria from the network consequently leading to their clearance through mitophagy [38].

Defects in mitochondrial quality control impair the renovation of the mitochondrial pool and exacerbate mitochondrial and cellular damage after liver I/R (Figure 1) [39,40]. Nutrient and oxygen deprivation during the ischemic period disrupts cellular respiration interfering with ATP synthesis. Paradoxically, reoxygenation further aggravates mitochondrial function due to accumulation of free radicals [41] and excessive uptake of Ca^2+^ into mitochondria [42]. Ca^2+^ participate in vast biological pathways as important signaling molecules [43]. The constant dynamic state of mitochondria makes them important organelles in the maintenance of cellular Ca^2+^ homeostasis, being able to uptake high quantities of these molecules. However, the excessive accumulation of Ca^2+^ in mitochondria during I/R results in the collapse of mitochondrial membrane potential. Together with decreased mitochondrial biogenesis and compromised mitochondrial dynamics, these factors trigger the permanent opening of MPT pores that results in mitochondrial depolarization and swelling [23,40,44]. Consequently, the onset of MPT induces cell death by apoptosis and necrosis [45].

### 3.1. Mitochondrial Biogenesis

The dynamic nature of mitochondria allows the reshaping of the mitochondrial network in adaptation to stresses of different natures. Mitochondrial biogenesis and mitochondrial dynamics are governed by a sophisticated signaling network involving the activation of many transcription factors in response to intracellular signals and environmental stimuli, such as nutrients and oxygen availability, growth factors or toxins [35]. Peroxisome proliferator-activated receptor (PPAR) gamma coactivator 1-alpha (PGC-1α) is the master regulator of energy metabolism being involved in many biological pathways, such as mitochondrial biogenesis, adaptive thermogenesis and glucose and fatty acid metabolism [46,47]. Activation of PGC-1α and its downstream transcription factors, including nuclear respiratory factors (NRF1 and NRF2) and mitochondrial transcription factor A (TFAM), stimulates the transcription and replication of mtDNA. mtDNA encodes multiple subunits of the mitochondrial respiration chain complexes and therefore is essential for the genesis of new mitochondria [48,49].

**Figure 1 biomedicines-11-00948-f001:**
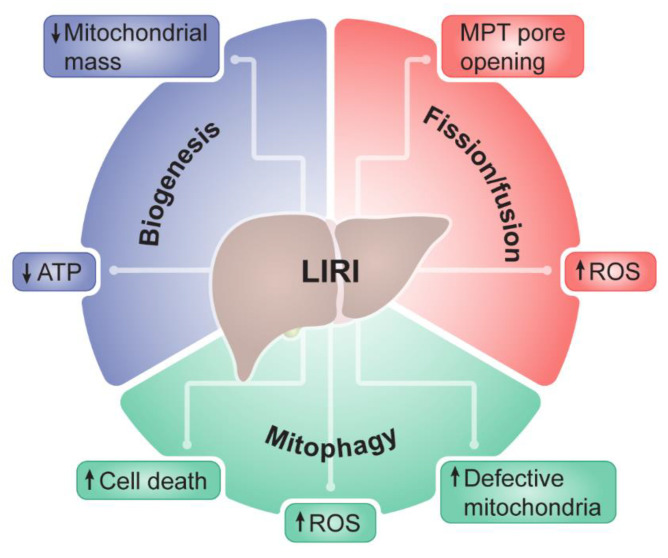
Mitochondrial quality control during liver ischemia/reperfusion injury (LIRI). Disruption of mitochondrial quality control processes, such as mitochondrial biogenesis, mitochondrial fission/fusion, and mitophagy, leads to mitochondrial dysfunction and is believed to exacerbate of LIRI.

Mitochondrial damage is one of the main causes for LIRI [8,50,51,52]. Studies report that both mitochondrial number and mtDNA content decrease during liver I/R [50,53], suggesting that both mitochondrial function and mitochondrial quality control mechanisms are compromised. In fact, mitochondria from rats subjected to liver transplantation were characterized by having poor bioenergetics and collapsed membrane potential [54]. As expected, a compromised oxidative phosphorylation, resulting in decreased energy production and altered hepatocyte morphology, led to the onset of LIRI. Pretreating the animals with berberine was sufficient to preserve mitochondrial function and to prevent tissue injury [54] (Table 1). Berberine is a natural compound with promising mitochondrial protective effects. Berberine prevents and reverts mitochondrial dysfunction by stimulating mitochondrial biogenesis through activation of the sirtuin 1 (SIRT1)-AMP-activated protein kinase (AMPK) axis [55,56]. During I/R, preconditioning with berberine increased the content of mitochondrial biogenesis-related proteins, such as PGC-1α, SIRT1 and SIRT3 [54]. Similarly, activation of heme oxygenase-1 (HO-1) in mice previous to I/R attenuated LIRI through improvement of mitochondrial function and regulation of mitochondrial quality control [57,58]. Maintenance of a mitochondrial healthy pool by activation of HO-1 using either cilostazol or hemin involved the activation of PGC-1α, NRF-1 and TFAM, which was associated with the increase of mtDNA copy number and mitochondrial mass. In fact, proteins relevant to mitochondrial biogenesis were found to be downregulated during I/R [50,53]. Activation of the SIRT1-AMPK pathway by treating mice with genipin or the SIRT1 activator, SRT1720, was effective in protecting mice against LIRI [50,53].

The energy sensing protein AMPK plays an important role in the regulation of mitochondrial homeostasis and metabolism. Under conditions of energy stress, AMPK is activated to, simultaneously, reduce energy consumption and increase ATP synthesis via stimulation of catabolic pathways [75]. To preserve mitochondrial health, AMPK activates the downstream transcription factor PGC-1α, in a process which requires SIRT1 and which stimulates mitochondrial biogenesis, dynamics, and quality [76]. It is thought that inducing the AMPK-PGC-1α signaling pathway protects mice livers against LIRI [59], since AMPK has protective effects against inflammation, oxidative stress, and apoptosis, which are all underlying pathological causes of I/R [59,77]. Evidence suggests that SIRT1 is necessary for the activation of AMPK through a positive feedback cycle [76]. Given its role in stress responses and its relevance in mitochondria [78], it is plausible to think that it has an important role in LIRI. Indeed, prolonged ischemia causes SIRT1 depletion in hepatocytes and its levels are not restored after reperfusion. Consequently, loss of SIRT1 leads to mitochondrial dysfunction and necrosis after I/R [60]. Activation of SIRT1 during I/R improved the clearance of defective mitochondria through mitophagy and led to the suppression of MPT and necrosis thereby attenuating LIRI [60]. Similarly, increasing SIRT1 levels protected fatty livers against LIRI by decreasing oxidative stress, activating AMPK, and suppressing apoptosis [79]. Curiously, mitofusin-2 (MFN2), a key protein for mitochondrial fusion, was found to have an important role after LIRI. Silencing MFN2 in cells overexpressing SIRT1 abolished the cytoprotective effect of SIRT1 [60].

### 3.2. Mitochondrial Fission and Fusion

Mitochondrial turnover is crucial for cellular wellbeing. To prevent mitochondrial dysfunction and cellular damage, cells rely on the permanent balance between mitochondrial biogenesis, dynamics and mitophagy. Mitochondrial networks adapt to various cytosolic signals by transitioning between different states. Mitochondria become energetically more efficient when in a hyperfused state whereas clearance of dysfunctional organelles is favored when in a microfused state [80,81]. Transition between these two states requires events of mitochondrial fission and fusion. Upon post-translational modification of dynamin-related protein 1 (DRP1) in the cytosol, DRP1 is translocated to the outer mitochondrial membrane (OMM) where it interacts with its receptors mitochondrial fission factor (MFF), mitochondrial dynamic proteins of 49 kDa (MID49), MID51, and mitochondrial fission 1 protein (FIS1) [81,82,83]. The subsequent oligomerization of DRP1 leads to the constriction and scission of mitochondria. Whereas mitochondrial fusion is initiated with the association between molecules of MFN1 that are anchored to the OMM of the two organelles, the interaction between optic atrophy protein 1 (OPA1) and cardiolipin promotes the fusion of the IMM [81,82,83]. It is believed that MFN2 has a similar role to MFN1 in mitochondrial function, however it has not been fully elucidated yet [81,83].

A growing body of evidence supports the suggestion that I/R injury is influenced by mitochondrial dynamics [63,84,85,86]. Excessive mitochondrial fragmentation is thought to increase cell susceptibility to the MPT onset [87] and to trigger cell death by apoptosis [88], which are hallmarks of LIRI. Reducing mitochondrial fission suppressed the MPT pore opening and decreased cell death after cardiac I/R injury [89]. Moreover, preserving mitochondrial function and suppressing apoptosis during I/R protected mice livers against LIRI [61] (Table 1). Studies have reported that the content of mitochondrial fission- and fusion-related proteins including DRP1 and MFN2 is altered with liver I/R, suggesting that mitochondrial dynamics mechanisms are imbalanced during liver I/R [50,58]. Therefore, it is reasonable to assume that manipulation of mitochondrial fragmentation by directly targeting fission and fusion during liver I/R might have protective effects against LIRI. Indeed, decreasing DRP1 protein levels reduced mitochondrial fission and attenuated LIRI [40,50,58]. Similarly, administration of the hormone irisin protected mice against LIRI by inhibiting DRP1 and FIS1, increasing mitochondrial biogenesis and suppressing apoptosis [40]. Overexpression of augmenter of liver regeneration (ALR) in mice granted resistance over LIRI [61,62]. ALR suppressed the phosphorylation and translocation of DRP1 to the OMM in a mechanism involving cyclin-dependent kinase 1 (CDK1) and cyclin B [62]. Normal activation of DRP1 is regulated by post-translational modifications such as phosphorylation and SUMOylation [90,91]. Curiously, SUMOylation of DRP1 was also found to be controlled by ALR [63]. Manipulation of DRP1 SUMOylation through ALR protected hepatocytes from mitochondrial fragmentation and injury after I/R [63]. Furthermore, decreasing the levels of *Drp1* methylated mRNA and its translation impaired DRP1-related mitochondrial fission and ameliorated LIRI [64]. Further evidence suggests that regulation of mitochondrial fusion may also have beneficial effects after I/R. Knocking-out toll-like receptor 4 (TLR4), a protein involved in inflammation, resulted in the upregulation of MFN2 and PGC-1α, and thus improving mitochondrial function and reducing LIRI [65]. However, amelioration of LIRI was also verified by treating mice with silibinin which decreased the levels of mitochondrial fusion-related proteins MFN1 and OPA1 [66].

### 3.3. Mitophagy

Cell survival depends on the quality of the mitochondrial pool. Accumulation of damaged mitochondria have dire consequences for a cell’s fate. Excessive increase in mitochondrial Ca^2+^ can lead to mitochondrial dysfunction [92,93]. High levels of Ca^2+^ induce the MPT, leading to mitochondrial swelling and triggering the release of cytochrome *c*, which result in apoptotic death [94]. Thus, cells must ensure proper elimination of defective mitochondria. The removal and recycling of unwanted organelles is accomplished through autophagic processes that include macroautophagy, microautophagy, and chaperone-mediated autophagy. In macroautophagy, the major class of autophagy, cellular material is engulfed in autophagosomes that then fuse with lysosomes. The enclosed material is promptly degraded by hydrolytic enzymes in autolysosomes [95]. The selective removal of mitochondria through macroautophagy is denominated mitophagy. Mitophagy ensures adequate cellular homeostasis as its inhibition results in the collapse of mitochondrial function contributing to several human diseases such as metabolic disorders, Alzheimer’s disease, cancer and aging [96,97]. Two main regulatory pathways are responsible for the regulation of mitophagy in mammals, ubiquitin-mediated mitophagy and receptor-mediated mitophagy. These pathways will not be covered here in depth as it is beyond the scope of the current review and they have been extensively reviewed elsewhere [96,98]. Briefly, ubiquitin-mediated mitophagy is dependent upon the interaction between phosphatase and tensin homologue-induced putative kinase 1 (PINK1) and Parkin. Under basal conditions, PINK1 is quickly degraded by ubiquitin-proteasome system after being recruited to the IMM [99]. However, depolarization of mitochondria hinders mitochondrial import and so PINK1 remains stabilized in the OMM [100]. Autophosphorylation of PINK1 leads to the recruitment and activation of Parkin in damaged mitochondria [101]. Substrates of Parkin such as MFN, Miro, and voltage-dependent anion channel (VDAC) are poly-ubiquitinated due to its E3 ligase activity being recognized by adaptor proteins that are anchored to microtubule-associated protein 1A/1B light chain 3 (LC3) proteins in the phagophore [98]. Clearance of mitochondria through receptor-mediated mitophagy involves proteins such as BCL2 interacting protein 3 (BNIP3), BCL2 interacting protein 3 like (NIX) and FUN14 domain containing 1 (FUNDC1) [96,98]. These mitophagy receptors are localized in the OMM and interact directly with LC3 to initiate mitochondrial engulfment and degradation.

Multiple types of cell death contribute to the development of liver damage following liver I/R. Necrosis is the most predominant, although programmed cell death pathways such as apoptosis, ferroptosis, and pyroptosis are also implicated [4,102,103]. Autophagy is typically associated with cell death pathways but it can also act as a mediator of apoptosis, or it can even occur independently (autophagy-dependent cell death) [104]. Nutrient starvation is a trigger of autophagy via regulation of canonical AMPK and mammalian target of rapamycin (mTOR) complex 1 (mTORC1) signaling pathways [105]. In brief, activation of AMPK inhibits mTORC1 which results in the activation of Unc-51 like autophagy activating kinase (ULK1), marking the initiation of autophagy [106,107]. The formation of the autophagosome is characterized by the sequential assembly of protein complexes that involve the recruitment and activation of proteins as beclin-1 (BECN1), ATG7 and LC3 [108]. The mechanisms underlying autophagy during and after liver I/R have been a topic of great debate for the past years as they appear to have a duality of effects [109]. Which possibly results from the employment of different I/R models, utilization of different treatments, and different conditions [103,109,110]. Nonetheless, manipulation of autophagy remains an appealing therapeutic that warrants further studies to clarify the complex mechanisms underlying it during LIRI. Moreover, the enhancement of mitophagy during I/R has shown promising effects [50,67,69]. It is believed that despite nutrient depletion during prolonged ischemia being a trigger of autophagy, the energy deficiency that accompanies it is a major factor in its impairment [23,111]. Loss of ATP is likely to halt autophagosome formation and thus to impair autophagy and mitophagy [23,112]. Similarly, in an advanced stage of reperfusion the accumulation of Ca^2+^ in mitochondria leads to activation of calpains and in turn proteins implicated in the autophagy machinery, such as ATG7 and BECN1, are hydrolyzed [112]. Energy insufficiency in association with the degradation of autophagy-related proteins contributes to a vicious cycle in which failure to remove deficient mitochondria leads to ROS accumulation, uncoupling of oxidative phosphorylation and cell death. Exploring mitophagy as a therapeutic approach holds great promise not only in diverse pathological conditions but also in aging [97]. Indeed, mitophagy modulators such as urolithin A and spermidine have been shown to have therapeutical potential against physiological decline [113,114].

Mitophagy-mediated elimination of unhealthy mitochondria is expected to enhance mitochondrial and cellular health by impairing the onset of the MPT and improving mitochondrial function, thereby preventing hepatocellular death after LIRI. An increasing number of studies report that mitophagy is disturbed during I/R. After I/R both autophagy- and mitophagy-related proteins are dysregulated, which is likely to be correlated with impaired mitochondrial function, cell death, and progression of LIRI. Although the mechanisms of mitophagy during I/R are still not fully understood, modulation of mitophagy seems to have beneficial effects for the amelioration of LIRI (Table 1). Treatment of mice with genipin before I/R is suspected to restore ubiquitin-dependent mitophagy after I/R as shown by the increase in mitochondrial Parkin [50]. SIRT1 and AMPK are likely to be interconnected and to play a relevant role in the regulation of mitophagy during I/R since inhibition of SIRT1 decreased phosphorylation of AMPK and impaired the effects of genipin on LIRI [50]. I/R injury is aggravated in alcoholic fatty liver-induced mice being characterized by impaired autophagy and mitophagy, increased inflammatory response and extensive hepatocellular death. Pretreatment with 2-methoxyestradiol restored the expression of autophagy-related proteins including BECN1, ATG5, ATG7, LC3-II, and Parkin. This was found to attenuate LIRI through a SIRT1-dependent mechanism [67]. Actually, SIRT1 is also thought to induce mitochondrial autophagy through direct deacetylation of MFN2 [111]. In a recent study, stimulation of mitophagy using 5-aminoimidazole-4-carboxamide ribonucleotide (AICAR), an AMPK activator, during I/R in diabetic mice improved an already aggravated I/R injury. Oxidative stress was attenuated and PINK1/Parkin-related mitophagy was stimulated [68]. Besides being deeply involved in the regulation of energy metabolism, AMPK also regulates a variety of aspects crucial for mitochondrial homeostasis. It is known to regulate mitochondrial biogenesis, dynamics and mitophagy [75]. Its participation in these processes was found to be crucial for its role in the regulation of I/R injury [115,116]. However, the signaling pathways by which it contributes to the amelioration of LIRI are still being uncovered. Additional evidence supports the role of AMPKα in the alleviation of LIRI [69]. Transfusion of mesenchymal stem cells into mice after reperfusion attenuated I/R-induced hepatocellular damage through activation of AMPKα-mediated PINK1-dependent mitophagy [69]. Similarly, both 25-hydroxycholesterol and pterostilbene improved LIRI via activation of PINK1-mediated mitophagy [70,71]. Gu and colleagues provided evidence suggesting that PINK1 is translocated to mitochondria upon I/R through mitochondria-associated membranes, and knocking down PINK1 aggravates LIRI by increasing the pool of dysfunctional mitochondria [117]. Moreover, modulation of Parkin also seems to be beneficial for I/R injury. Treatment of mice with resolvin D1 improved mitochondrial health by reducing mitochondrial swelling, decreasing oxidative stress, and modifying mitochondrial-related proteins including PGC1α, NRF1, TFAM, DRP1, PINK1 and Parkin. Additionally, it was found that the observed effects were mediated by thioredoxin 2 (TRX2)-thioredoxin-interacting protein (TXNIP) signaling which is being discovered as a regulator of mitochondrial quality [72]. Parkin deficiency exacerbates LIRI in rats by suppressing mitophagy [118]. Other pathways of mitophagy are also involved in I/R. For instance, Zhou and colleagues demonstrated that CCAAT/enhancer-binding protein homologous protein (CHOP) knockout decreased hepatocellular death during LIRI. The absence of CHOP led to upregulation of DRP1 and BECN1 and thus increased mitophagy following I/R in mice [73].

The synergy between improved mitochondrial biogenesis and mitophagy resulted in an increased cellular tolerance against LIRI. Mitophagy is interconnected with mitochondrial dynamics as evidenced by Kong and colleagues, where silencing *Mfn2* impaired the effects of ALR during LIRI primarily by impairing ubiquitin-dependent mitophagy [74]. However, most previous studies share a common limitation. The lack of a methodology to directly analyze mitophagy has led to the utilization of indirect methods, such as measuring the content of autophagy- and mitophagy-related proteins.

## 4. Impact of Liver Conditions on LIRI

The increasing demand for liver transplants has led to an alarming number of patients on waiting lists [1,2]. In spite of significant improvements in the past few decades that have improved the success of liver transplantation and improve the quality of life of patients post-surgery, many patients with end-stage liver disease die while waiting for transplantation [119]. Inclusion of marginal liver donors in the liver donor pool is a plausible strategy to neutralize these effects [2]. Currently, liver steatosis and advanced donor age are among the main reasons to discard available livers due to their higher susceptibility to I/R injury that contributes to post-surgery complications and lower survival rate. Decreasing the detrimental effects of I/R injury in livers from marginal donors would increase the number of donors and improve the outcome of liver surgery. The development of such strategies is dependent on the understanding of the underlying mechanisms of I/R injury in livers from marginal donors. In the following sections, we will discuss the impact of age and steatosis in liver transplantation and possible strategies to minimize liver susceptibly to I/R injury (Table 2).

### 4.1. Fatty Liver Disease

Obesity is a major concern for human health, especially in developed countries, as it has been on the rise for many decades and is associated with many comorbidities. Obesity increases the risk of developing several diseases including metabolic syndrome, type 2 diabetes (T2DM), cardiovascular diseases, and non-alcoholic fatty liver disease (NAFLD) [129]. NAFLD is characterized by the accumulation of triglycerides in hepatocytes (hepatic steatosis) that prompts hepatocyte injury and death, as well as liver inflammation [130]. The progression of NAFLD is related to cirrhosis and hepatocellular carcinoma (HCC). Hepatic steatosis is typically associated with obesity, as substantial accumulation of body fat results in the dysfunction and death of adipocytes [131]. In turn, development of insulin resistance due to secretion of cytokines and inflammatory mediators by adipocytes leads to the release and accumulation of free fatty acids in hepatocytes [130]. This triggers the synthesis of triglycerides by hepatocytes which is associated with the accumulation of diacylglycerols intermediates that lead to the development of hyperglycemia [130]. The excessive accumulation of free fatty acids in the liver makes it more susceptible to injury due to mitochondrial uncoupling and overproduction of ROS through the mitochondrial respiratory chain, stimulation of endoplasmic reticulum stress and activation of cell death receptors [130,131]. Interestingly, mitochondria from injured hepatocytes release DAMPs that activate HSCs and stimulate the progression into liver fibrosis [132]. Liver steatosis increases the risk of morbidity and mortality after liver surgery including liver transplantation and resection [133]. NAFLD is a major risk factor for increased HCC recurrence after liver transplantation [134,135]. Liver transplantation is associated with increased recurrence of HCC, which is likely associated with LIRI [136,137]. The release of proinflammatory cytokines, growth factors and ROS, as well as changes in the liver microenvironment, facilitate the formation and development of metastasis after liver transplantation [136,138,139,140]. While the role of mitochondria in HCC recurrence after liver transplantation is not well understood, it can be assumed that mitochondrial function and health are important contributors to HCC progression. Poorly regulated mitochondrial metabolism, dynamics, ROS, and mitophagy exacerbate proliferation and growth of HCC cells and the development of metastasis [141,142].

Steatotic livers are more vulnerable to I/R injury than non-steatotic livers. In fact, the mechanisms underlying LIRI seem to be different for both cases [26,143]. Apoptosis was shown to be the prevalent type of cell death after I/R injury in normal rat livers while necrosis was the main type of death following I/R injury in steatotic rat livers [143]. Moreover, since mitochondrial dysfunction is a major contributor to the development and progression of NAFLD [144], it might be a likely cause for the increased susceptibility of steatotic livers to I/R injury. During NAFLD, loss of hepatic mitochondrial homeostasis leads to increased oxidative stress that impairs mitochondrial respiration, causes lipid peroxidation, increases the secretion of cytokines, and eventually leads to hepatocyte death [145]. Indeed, higher generation of superoxide was observed during reperfusion in fatty rat livers than in lean rat livers, which is thought to be linked to increased lipid peroxidation and higher susceptibility to LIRI [41]. The superoxide anion is a main type of ROS that when produced in excessive quantity is typically associated with damage to mtDNA and mitochondrial-encoded respiratory chain proteins, thus deteriorating mitochondrial function [146]. ROS-mediated damage to mitochondrial respiratory complexes may help explain the impairment of mitochondrial energy metabolism after I/R in steatotic livers. Caraceni and colleagues demonstrated that oxidative phosphorylation is significantly impaired in fatty livers [147]. In this study, complex I and ATPase activities were found to be decreased during I/R and, after reperfusion, ATP levels were not recovered in steatotic livers [147]. These results are consistent with a later study where the authors show that warm I/R leads to mitochondrial complex I dysfunction and consequently to higher susceptibility of steatotic livers to LIRI [148]. Complex I is a mitochondrial site that is associated with significant ROS production [149]. Mitochondrial uncoupling protein 2 (UCP2) regulates metabolism by working as an uncoupler of oxidative phosphorylation from ATP generation [150]. Interestingly, UCP2 deficiency ameliorated LIRI and increased the survival of mice after I/R [120]. UCP2 was found to be responsible for hepatocyte susceptibility to hypoxia/reoxygenation by reducing mitochondrial membrane potential and ATP levels [121]. Moreover, activation of the AMPK-SIRT1 signaling pathway by renalase mitigated oxidative stress and alleviated mitochondrial dysfunction in steatotic livers [122]. Induction of HO-1 in steatotic livers has also shown promise in the amelioration of steatotic liver I/R injury [151]. Stimulation of mitochondrial function to decrease steatosis-induced susceptibility to LIRI may be a promising strategy to increase the use of marginal liver donors in liver transplantation.

### 4.2. Aging

Organismal function progressively declines during adulthood, eventually resulting in death. For a long time, researchers have been looking for approaches to hinder this functional decline to increase the health span and life expectancy of human beings. Current efforts are focused on understanding the driving causes of aging and in the discovery of anti-aging interventions, including nutrient restriction and natural compounds [152,153,154]. The causes of aging are currently summarized in well accepted hallmarks [155,156]. Mitochondrial and metabolic dysfunction have major roles in aging [7], since age progression is accompanied by metabolic changes, decline of mitochondrial function, and decline of autophagy efficiency [157,158,159]. The liver is a major regulator of systemic metabolism. But despite having a notable resilience to aging its function still declines during aging, leading to an increase of the incidence of liver diseases [160,161,162]. The decrease of its function is correlated with impaired mitochondrial bioenergetics and increased oxidative injury caused by mtDNA mutations, oxidative stress, and defects in oxidative phosphorylation. Genomic instability, telomere attrition, epigenetic alterations, and deregulated nutrient sensing pathways are some of the other causes underlying age-related liver dysfunction [161]. Evidence suggests that surgical interventions including liver transplantation, as well as resection, have worse outcomes in aged patients than in younger patients [163,164]. Using aged liver grafts for liver transplantation is associated with higher mortality of the recipient [162,165,166]. However, this is source of controversy as systematic reviews demonstrate that the impact of age on liver interventions is not significant, ruling out age as an exclusion criterion for liver resection [167,168]. However, more studies are warranted to clarify if this controversy simply arises from careful liver donor selection or from the type of disease that the patients have at the time of liver resection [169,170,171]. Nonetheless, the increased morbidity and mortality associated with the outcome of liver transplantation is in part due to sensitivity to LIRI. Aged livers have been associated with higher susceptibility to I/R injury in the context of liver transplantation and resection [172,173,174]. In comparison to young mice, adult mice had increased liver injury in response to I/R injury, which was shown to be associated with altered inflammatory response and impaired neutrophil function [174]. Similarly, livers from 9 month-old rats that underwent warm I/R had increased hepatocellular injury compared to livers from 2 month-old rats [173]. Following I/R, livers from old animals were characterized by having increased cellular stress, reduced vasodilation and sinusoidal capillarization [123]. Simvastatin inhibits hydroxy-methylglutaryl coenzyme A (HMG-CoA) and is typically used to lower lipid blood levels. Its administration before I/R in old animals protected their livers against damage [123].

Aging is characterized by the progressive loss of mitochondrial function. With aging the mitochondrial morphology is altered [175], the efficiency of oxidative phosphorylation declines [176], and mitochondrial quality control mechanisms are impaired [159]. Since mitochondrial dysfunction has a big impact on liver’s susceptibility to LIRI, it is possible that aged livers might be more vulnerable to I/R injury in part because of age-related mitochondrial dysfunction. Indeed, Selzner and colleagues noticed that hepatic ATP content decreases after ischemia in both young and old mice. However, contrary to young livers, ATP content is not recovered after reperfusion in the livers of old mice [124]. These results hint that the impact of old age in oxidative phosphorylation and consequently on ATP synthesis exacerbates LIRI. Preconditioning the livers of old mice with short periods of I/R and glucose administration before ischemia was enough to increase ATP content after reperfusion and protect against injury [124]. I/R injury was found to particularly aggravate mitochondrial function in old mice as demonstrated by depletion of proteins relevant to maintain mitochondrial quality such as SIRT1 and MFN2 due to increased expression of calpains [125]. Overexpression of SIRT1 and MFN2 reduced I/R injury in old mice by preventing the opening of the MPT pore, stimulating mitophagy and promoting mitochondrial function [125]. During I/R there is a decrease of calpastatin, which is an inhibitor of calpains. This decrease is aggravated in old animals resulting in mitochondrial dysfunction, impaired autophagy, and hepatocyte death. Its overexpression was shown to be protective against LIRI [126]. The accumulation of unhealthy mitochondria during aging is likely associated with a decline in mitophagy [97]. Age-associated defects in mitophagy were found to exacerbate LIRI in mice as proteins that play essential roles in mitophagy and autophagy, such as Parkin and ATG5, were found to be decreased in the livers of old mice [177]. Actually, improving autophagy by administration of rapamycin and ischemic preconditioning successfully protected livers from I/R injury in old mice [127]. Rapamycin-mediated inhibition of mTOR was shown to improve mitophagy [178], and to have positive effects on lifespan extension [179]. There are several strategies that are currently being study to reverse aging. Parabiosis consists in uniting two living animals resulting in the sharing of blood supply between the two [180]. Circulating factors present in the blood of young mice are thought to have rejuvenating effects and has shown promising effects on restoring synaptic plasticity and improving cognitive function in old mice [181]. Interestingly, administration of plasma from young mice to old mice before I/R attenuated LIRI by improving autophagic activity due to activation of the AMPK/ULK1 signaling pathway [128]. With age progression there is dysregulation of the immune system resulting in a higher tendency for the development of inflammation [182]. Mitochondrial dysfunction and activation of the nucleotide-binding domain and leucine-rich repeat containing protein 3 (NLPR3) inflammasome are potential mechanisms for age-associated inflammation. Since one of the causes for LIRI is related to the inflammatory immune response, its suppression might be a promising strategy to ameliorate LIRI in old livers. Recently, it was found that hepatocytes from old mice release more mtDNA than those from young mice after I/R [183]. Circulating mtDNA has been recognized has being a proinflammatory DAMP [4]. Indeed, increased release of mtDNA activated the NLPR3 inflammasome leading to the stimulation of a macrophage-mediated proinflammatory response that exacerbated LIRI [183].

## 5. Concluding Remarks

This review provides current knowledge on mitochondrial quality control during LIRI and proposes that the preservation of mitochondrial health is essential for improving patient outcomes. Our knowledge of liver I/R has increased significantly over the last few decades. Understanding the complex mechanisms underlying liver I/R is crucial for improving the clinical success of liver surgeries such as liver transplantation and resection. The ubiquity of mitochondria, as shown by their involvement in several metabolic and signaling pathways, makes them extremely relevant organelles for the maintenance of whole-body homeostasis. Dysregulation of mitochondrial function contributes to the development and progression of many human diseases. As anticipated, mitochondria are also important for liver I/R injury. The production of mitochondrial ROS, calcium overload, and increased mitochondrial permeability are hallmarks of LIRI. These hallmarks are being targeted by emerging strategies that focus on improving mitochondrial structure and function through modulation of mitochondrial biogenesis and dynamics, as well as mitophagy. Thus, a better understanding of mitochondrial and metabolic functions will allow the development of novel interventions aimed at ameliorating LIRI in livers from healthy and marginal donors, including steatotic and old livers. For instance, potential interventions may include supplementation of preservation solutions with chemical compounds or even administration of chemical compounds during reperfusion of the liver to increase the success of the transplanted grafts.

Interestingly, the release of DAMPs from damaged mitochondria into the circulation of liver donors correlates with early graft dysfunction in liver transplant recipients [184,185]. Dysregulation of mitochondrial function is implicated in graft dysfunction and rejection after liver transplantation [186]. Mitochondrial injury is closely related to the outcome of liver transplantation. Indeed, improving mitochondrial function with machine perfusion leads to improved graft function [8,187]. Machine perfusion allows the monitoring of mitochondrial function and the detection of mitochondrial-related biomarkers [187,188]. Both parameters could help predict liver function before liver implantation and liver morbidity and mortality after transplantation [187,188]. A clinical study found that poorly performing mitochondria from human patients were correlated with worse complications after liver transplantation [189].

## Figures and Tables

**Table 1 biomedicines-11-00948-t001:** Interventions that ameliorate or protect against liver ischemia/reperfusion injury (LIRI) by interfering with mitochondrial quality control mechanisms.

Intervention	Species/Model	Mechanism	Effect on Mitochondria	Reference
Mitochondrial biogenesis				
Berberine	Wistar rats	–	↑ ATP content ↓ ROS ↑ Mitochondrial biogenesis markers	Martins et al. [54]
Cilostazol	Mice HepG2 cells	Activation of HO-1 and Nrf2	↑ Mitochondrial biogenesis markers ↑ Mitochondrial function	Joe et al. [57]
Hemin	Mice	Activation of HO-1	↑ Mitochondrial biogenesis markers ↑ Fission/fusion markers ↑ Mitophagy	Hong et al. [58]
Genipin	Mice	Activation of AMPK and SIRT1	↑ Mitochondrial biogenesis markers ↑ Fission/fusion markers ↑ Mitophagy	Shin et al. [50]
SRT1720	Mice	Activation of SIRT1	↑ Mitochondrial biogenesis markers ↑ Mitochondrial function ↑ Mitochondrial mass	Khader et al. [53]
Isolongifolene	Mice	Activation of AMPK and PGC1α	↓ Oxidative stress	Li et al. [59]
*Sirt1* overexpression	Mice (Ad-SIRT1) Primary hepatocytes	Interaction between SIRT1 and MFN2	↓ MPT ↓ Mitochondrial dysfunction	Biel et al. [60]
Mitochondrial fission/fusion				
*Alr* overexpression	Mice (Ad-HSS) BEL-7402 cells (Ad-HSS)	–	↑ Mitochondrial function ↓ Mitochondrial ROS ↓ Mitochondrial-related apoptosis	Jiang et al. [61]
*Alr* overexpression	HSS^+/–^ mice HepG2 cells	Translocation and activation of DRP1	↑ Fission markers	Zhang et al. [62]
Irisin	Mice HL-7702 cells	Inhibition of excessive fission (through inhibition of DRP1 and FIS1)	↑ Mitochondrial biogenesis markers ↑ Mitochondrial content ↓ Oxidative stress	Bi et al. [40]
*Alr* silencing	ALR^+/–^ mice	DRP1 SUMOylation and recruitment to mitochondria	↓ Mitochondrial fission	Huang et al. [63]
*Fto* overexpression	Mice (Ad-FTO)	Inhibition of DRP1	↓ Mitochondrial fragmentation ↓ Oxidative stress	Du et al. [64]
*Tlr4* silencing	TLR4-KO mice	Activation of IL6 and TNFα pathways	↑ Mitochondrial biogenesis markers ↑ Mitochondrial fusion markers ↓ ROS	Zhang et al. [65]
Silibinin	Wistar rats	–	↑ Mitochondrial fusion markers	Qajari et al. [66]
Mitophagy				
2-Methoxyestradiol	AFL mice	Activation of SIRT1	↑ Mitophagy	Cho et al. [67]
AICAR	*db*/*db* mice		↑ Mitophagy	Zhijun et al. [68]
UC-MSC transfusion	Mice L02 hepatocytes	Activation of AMPKα	↓ Mitochondrial ROS ↑ Mitophagy	Zheng et al. [69]
Pterostilbene	Mice L02 hepatocytes	Activation of PINK1	↓ Mitochondrial dysfunction ↓ Mitochondrial ROS ↑ Mitophagy	Shi et al. [70]
25-Hydroxycholesterol	Sprague Dawley rats	Activation of PINK1/Parkin pathway	↑ Mitophagy	Cao et al. [71]
Resolvin D1	Mice	Activation of TRX2	↑ Mitophagy markers ↓ Mitochondrial swelling ↓ Oxidative stress ↑ Mitochondrial biogenesis markers ↑ Mitochondrial fission markers	Kang et al. [72]
*CHOP* silencing	CHOP-KO mice	Activation of DRP1-Beclin1 pathway	↓ ROS ↑ Mitophagy	Zhou et al. [73]
*Alr* overexpression	Brown-Norway rats (Ad-ALR)	Activation of MFN2	↑ Mitochondrial function ↑ Mitophagy	Kong et al. [74]

↑, increase; ↓, decrease; Ad, adenovirus; AFL, alcoholic fatty liver; AICAR, 5-aminoimidazole-4-carboxamide ribonucleotide; ALR, HSS, augmenter of liver regeneration; AMPK, AMP-activated protein kinase; ATP, adenosine triphosphate; CHOP, CCAAT/enhancer-binding protein homologous protein; DRP1, dynamin-related protein 1; FIS1, mitochondrial fission 1 protein; FTO, fat mass and obesity associated; HO-1, heme oxygenase-1; IL6, interleukin 6; KO, knockout; MEF, mouse embryonic fibroblasts; MFN2, mitofusins 2; MPT, mitochondrial permeability transition; Nrf2, nuclear factor erythroid 2-related factor 2; PGC-1α, Peroxisome proliferator-activated receptor gamma coactivator 1-alpha; PINK1, phosphatase and tensin homologue-induced putative kinase 1; ROS, reactive oxygen species; SIRT1, sirtuin 1; TLR4, toll-like receptor 4; TNFα, tumor necrosis factor α; TRX2, thioredoxin 2; UC-MSC, umbilical cord-derived mesenchymal stem cell.

**Table 2 biomedicines-11-00948-t002:** Interventions that ameliorate or protect against liver ischemia/reperfusion injury (LIRI) during non-alcoholic fatty liver disease (NAFLD) and aging.

Intervention	Species/Model	Mechanism of Action	References
NAFLD			
*Ucp2* silencing	UCP2-KO *ob*/*ob* mice	Increased ATP levels Increased mice survival following I/R	Evans et al. [120]
*Ucp2* silencing	UCP2-KO primary hepatocytes	Increased cellular viability Increased ATP levels Increased mitochondrial membrane potential	Evans et al. [121]
Renalase	HFD mice HepG2	Increased NAD+ levels and activation of SIRT1 Decreased ROS production Increased mitochondrial function	Zhang et al. [122]
Aging			
Simvastatin	Wistar rats	Inhibition of HMG-CoA Decreased hepatocellular damage Decreased oxidative stress	Hide et al. [123]
Ischemic and glucose preconditioning	Mice	Increased ATP levels	Selzner et al. [124]
*Sirt1* and *Mfn2* overexpression	Mice Primary hepatocytes	Prevention of mitochondrial dysfunction Prevention of MPT pore opening Increased mitophagy Prevention of cell death	Chun et al. [125]
Calpastatin overexpression	Mice Primary hepatocytes	Mitochondrial elongation Prevention of MPT pore opening Prevention of mitochondrial depolarization Prevention of necrosis	Flores-Toro et al. [126]
Ischemic and rapamycin preconditioning	Mice	Increased autophagy	Jiang et al. [127]
Plasma from young mice	Sprague Dawley rats Primary hepatocytes	Activation of AMPK/ULK1 pathway Increased autophagy	Liu et al. [128]

AMPK, AMP-activated protein kinase; ATP, adenosine triphosphate; HFD, high fat diet; HMG-CoA, hydroxy-methylglutaryl coenzyme A; I/R, ischemia/reperfusion; KO, knockout; MFN2, mitofusin-2; MPT, mitochondrial permeability transition; NAD, nicotinamide adenine dinucleotide; NAFLD, non-alcoholic fatty liver disease; ROS, reactive oxygen species; SIRT1, sirtuin 1; UCP2, uncoupling protein 2; ULK1, Unc-51 like autophagy activating kinase.

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
