# Peer review of "Preservation of Mitochondrial Health in Liver Ischemia/Reperfusion Injury"

_biomedicines, 2023, doi:10.3390/biomedicines11030948_

Round 1
Reviewer 1 Report
This manuscript deals with "Preservation of mitochondrial health in liver ischemia/reperfusion injury". This article claims that mitochondrial health is very important for liver health . I suggest a minor correction and require a detailed clarification. Correction to be addressed by the authors as follows: The abstract is not well organized, where the sentences are incomplete and no continuity is there. It would be feasible, if include the significance of the current study in the abstract.
A brief description of how the authors selected information from the literature in the databases, as well as what time period they searched for, is missing.
Authors should justify and expand the information on the advantages and disadvantages of this synthetic material.
Authors should specify the main experimental conditions used on the evidences from the literature. Where they briefly describe the most important data reported in the literature in a homogeneous manner and sequence reinforcing the relevance of this agent as novel alternative.
Authors should discuss whether the use of this mitochondria targeting agents represents a solid alternative to existing commercial drugs. Also please discuss about the mitochondria on live cancer.
Please add schematic figure for mitochondrial targeting mechanism in liver
Please compare possible antioxidative effects of natural/synthetic agents on liver mitochondria.
Please add below studies to your manuscript in discussion section using below manuscripts:
DOI: 10.1155/2021/4946711
DOI: 10.1055/s-0042-110178
DOI: 10.1007/s12272-016-0766-0
Conclusions should reaffirm the fundamental contribution of this paper.
Reviewer 2 Report
A well written review which comprehensively accounts existing understanding on the role of mitochondria in isheamia/repererfusion injury during liver transplantation.
Reviewer 3 Report
The paper submitted by Machado et al reviews historical facts and updates related to the involvement of mitochondria in liver I/R and the impact of liver defects on LIRI. The ms is well written and easily readable in its descriptive part but needs to be more elaborated and to provide the personal touch of the authors on this topic. Indeed, the main information presented in this ms was previously published in review articles. This is supported by the fact that only 40% of the references presented are original research papers whereas 60% are reviews. The authors should also clearly discriminate human and non-human data to accurately and clearly evaluate the preclinical and clinical developments and their alignment.
Some additional comments are here detailed:
- Abstract: line 6 “their function…… to maintain” please add: cellular activity
- What about the compartmentalized effect of I/R that could concern hepatocytes from the different three zones (cholangiocytes in zone 1) of the liver parenchyma?
- The authors should clearly mention the liver cell types that will be addressed/reported (Hepatocytes, LSECs or others) in their current ms and provide the rationale of this selection. Any information reported on other liver cell types like stellate cells when NAFLD progresses to fibrosis for instance? How are mitochondria of the cells (myofibroblasts...) that will replace the dead hepatocytes and of those which infiltrate the altered liver parenchyma?
- Paragraph 2: the authors should add “cellular” in the corresponding title. A figure summarizing the cells and pathways documented so far should also be provided.
- Paragraph 3: Figure 1 should be more elaborated.
- It would be of interest if the authors describe the preclinical in vitro and in vivo models currently developed and used for investigating I/R and LIRI
- Paragraph 4: the authors should reorganize the section related to therapeutic strategies that have been addressed to modulate mitochondria activity (small molecules, herbs, cell and gene therapy…). A table recapitulating these data and the proposed mechanisms of action is mandatory
- we know from literature that alterations of the graft even after successful transplantation can be noticed at different times post-transplantation. Are mitochondria involved?
- Paragraph 5: this section should be rewritten with proposed perspectives/personal points of view of the authors
Reviewer 4 Report
A review by Machado et al. is devoted to the analysis of modern literature data on the role of mitochondria in the development of liver ischemia-reperfusion injury. The review discusses the mechanisms of maintaining mitochondrial homeostasis and disruption of these mechanisms in liver ischemia-reperfusion injury. The authors conclude that targeting mitochondria during I/R in steatotic and aged livers seems to decrease their inherent susceptibility to LIRI. This interesting review is well structured, concisely written.
Comments:
1. The review can be further illustrated with a figure or table. For example, a table could be added showing how drugs that affect LIRI affect mitochondrial biogenesis, mitophagy, mitochondrial dynamics, the MPTp, oxidative phosphorylation, and oxidative stress.
2. On page 6 section 3.3 the authors write: «Defective mitochondria accumulate high levels of Ca2+ causing mitochondrial swelling which then triggers the release of cytochrome c and consequently leads to apoptotic death». This is not a completely correct statement, since healthy mitochondria are also able to accumulate even more Ca2+ in relation to defective mitochondria.
Round 2
Reviewer 3 Report
The authors have satisfactorily addressed my concerns
few typo errors are to correct